# Synergistic Lanthanum-Cysteine Chelate and Corn Steep Liquor Mitigate Cadmium Toxicity in Chinese Cabbage via Physiological–Microbial Coordination

**DOI:** 10.3390/plants14193040

**Published:** 2025-10-01

**Authors:** Fengbo Ma, Zihao Wang, Wenhao Wang, Xian Wang, Xiaojing Ma, Xinjun Zhang, Yanli Liu, Qing Chen, Kangguo Mu

**Affiliations:** Beijing Key Laboratory of Farmyard Soil Pollution Prevention-Control and Remediation, College of Resources and Environmental Sciences, China Agricultural University, Beijing 100193, China

**Keywords:** Cd translocation, Chinese cabbage, corn steep liquor, lanthanum-cysteine chelate, rhizosphere microbiota

## Abstract

Cadmium (Cd) contamination of soil threatens agricultural productivity and food safety. In this study, a dual-component remediation strategy combining lanthanum-cysteine chelate (CLa) and corn steep liquor (CSL) was developed to alleviate Cd toxicity in Chinese cabbage (*Brassica rapa* subsp. *pekinensis*). CLa enhanced photosynthetic efficiency, antioxidant enzyme activity, and root viability, while reducing Cd translocation to shoots. In contrast, CSL acted primarily through organic nutrient supplementation, stimulating chlorophyll synthesis and promoting the growth of beneficial rhizosphere microbes. Notably, the combined treatment (CLCS) exhibited a synergistic effect, significantly enhancing biomass production, nutrient uptake, photosynthetic performance, and oxidative stress tolerance, while reducing Cd accumulation in plant tissues. Furthermore, CLCS optimized the soil microenvironment and microbiota composition, reinforcing plant resilience under Cd stress. This study offers a promising and cost-effective approach for mitigation of heavy metal stress and crop productivity improvement by coordinated plant–microbe–soil interactions.

## 1. Introduction

Heavy metal contamination of soil represents a global environmental challenge, with 14–17% of arable land affected by metallic pollutants worldwide [1,2]. Among these contaminants, cadmium (Cd) has emerged as a particularly pervasive threat due to its high mobility and bioaccumulation potential. In this context, China faces acute risks, as 19% of agricultural soils exceed national heavy metal quality standards, with Cd being the most prevalent contaminant [3,4]. Cd tends to accumulate in the edible parts of leafy vegetables, thereby entering the food chain and posing potential risks to human health [5]. Furthermore, Cd adversely affects leafy vegetables by disrupting cell division, damaging chloroplast structures, and inducing the accumulation of reactive oxygen species (ROS), which ultimately inhibits plant growth and development [6]. Chinese cabbage (*Brassica rapa* subsp. *pekinensis*), which is a widely consumed leafy vegetable across East Asia, is particularly susceptible to Cd stress. This highlights the urgent need for effective remediation strategies in Cd-contaminated agroecosystems to safeguard environmental sustainability and food security.

Plants have evolved three primary physiological mechanisms to counteract Cd toxicity: (i) activation of antioxidant systems to scavenge ROS [7,8]; (ii) biosynthesis of metal-chelating compounds (e.g., phytochelatins, metallothioneins) for Cd sequestration [9]; and (iii) regulation of metal transporter proteins (e.g., heavy metal ATPases, natural resistance associated macrophage proteins) to limit Cd translocation to shoots [10]. Recent studies have identified lanthanum (La), a redox-active rare earth element (REE), as a promising amendment due to its dual capacity to enhance antioxidant enzyme activities and coordinate with ligands under Cd stress [11,12,13]. However, the agricultural application of La is significantly limited by its rapid immobilization through phosphate precipitation in soils, leading to poor bioavailability and inconsistent field performance [14,15].

Chelating La with amino acids, particularly L-cysteine (Cys), presents a viable approach to enhance La bioavailability [16,17,18]. The sulfhydryl (–SH) and carboxyl (–COOH) groups of Cys form stable coordination complexes with La while simultaneously serving as essential plant nutrients [19,20,21,22]. Despite this advantage, the high production costs of rare earth–amino acid chelates impede their widespread adoption. To overcome this limitation, this study proposes a novel synergistic approach combining La-Cys chelates (CLa) with corn steep liquor (CSL), which is a cost-effective agricultural byproduct containing bioactive peptides (500–2000 Da) and organic acids [23]. Importantly, CSL has demonstrated enhanced Cd stress mitigation and plant growth promotion when used in combination with other amendments [24,25,26], suggesting potential synergistic effects with CLa.

As the world’s largest producer of La and as a pioneer in agricultural REE applications [27,28], China provides an ideal setting for this investigation. However, three critical knowledge gaps remain: (i) the structural characteristics of La-Cys chelates are poorly understood; (ii) the combined effects of CLa and CSL on Cd-contaminated systems have not been systematically evaluated; and (iii) the rhizosphere microbial community responses to this combined treatment are unknown. To address these gaps, this study aims to synthesize and characterize CLa using advanced spectroscopic techniques, assess the efficacy of the CLa-CSL combination in reducing Cd accumulation and improving physiological responses in Chinese cabbage, and elucidate the underlying stress mitigation mechanisms through integrated biochemical and rhizosphere microbial community analyses. This multidisciplinary approach will contribute to the development of sustainable phytoremediation strategies for Cd-polluted agricultural systems.

## 2. Results and Discussion

### 2.1. Characterization of CLa

The synthesis process of CLa is illustrated in Figure 1a. XRD analysis revealed definitive evidence of chelate formation, as demonstrated by significant peak position shifts in the CLa pattern (Figure 1b,c) compared to pure L-cysteine (18.24° → 20.86°, 24.6° → 26.18°, 31.42° → 34.04°). Notably, the appearance of new diffraction peaks corresponded to characteristic LaS crystallographic planes, confirming successful La coordination through sulfur bonding. Complementary FTIR spectroscopic analysis (Figure 1c) provided molecular-level evidence of chelation. The Cys spectrum exhibited characteristic absorption bands at: 1421 cm^−1^ (carboxylate symmetric stretching, –COOH), 1584 cm^−1^ (amino group bending, –NH_2_) and 2546 cm^−1^ (thiol stretching, –SH) [29]. Notably, the CLa spectrum showed marked attenuation of the –SH stretching vibration at 2546 cm^−1^ and significant reduction in –COOH band intensity at 1421 cm^−1^. These spectroscopic modifications demonstrate bidentate coordination of La through both –SH and –COOH functional groups, forming a stable cyclic chelate structure. This coordination geometry enhances La bioavailability while reducing its environmental mobility through strong binding to the organic ligand.

### 2.2. Growth Responses to Cd Stress and CLCS Treatment

Chinese cabbage, a predominantly leafy vegetable, exhibits high Cd accumulation capacity [30]. Cd stress disrupts cell wall stability, damages organelle structures, and impairs protein activity, leading to reduced plant growth and biomass, inhibited photosynthetic pigment synthesis, and impaired nutrient transportation [31,32]. Based on this study’s preliminary experiments, the optimal dosage of CLa was determined to be 2.5 mg kg^−1^ soil (Appendix A), as it significantly enhanced photosynthesis in Chinese cabbage, reduced soil DTPA-Cd, and decreased Cd accumulation in edible tissues. Therefore, this concentration was selected for the subsequent experiments. Detailed experimental procedures and results are provided in the Appendix A. This study demonstrated that CLCS application significantly enhanced the growth and biomass of Chinese cabbage (Figure 2a,b). The plant height, root length, fresh weight and dry weight increased by 6.8%, 31.9%, 21.3% and 7.2% under CLCS treatment, respectively, compared to CK (Figure 2c–f). This yield improvement induced by CLCS can be attributed to improved root structure, improved nutrient absorption, increased photosynthetic pigment content and higher photosynthetic rate. Roots are essential for absorbing water and nutrients from the soil and transporting them to the shoot, which plays a vital role in plant growth [33]. CLCS significantly increased the root length, surface area, diameter, volume, tip density, branching frequency, intersection complexity and activity by 31.9%, 78.2%, 9.8%, 100.6%, 41.8%, 68.3%, 65.2% and 39.8%, respectively, compared to CK (Figure 2d, Table 1). The root system is a major site of cytokinin secretion, thus, improved root architecture may enhance cytokinin production and further promote plant growth [34]. Moreover, improved root development facilitated nutrient uptake in Chinese cabbage subjected to Cd stress. By microscopic observation of Arabidopsis thaliana seedling roots cultured with LaCl_3_, Jiao et al. demonstrated that low concentrations of La can stimulate endocytosis in root cells, thereby facilitating enhanced nutrient acquisition by the root system [35]. In this study, the N, P, and K accumulation in Chinese cabbage increased by 21.2%, 11.3% and 33.6%, respectively, compared to CK (Figure 2g–i). This enhanced nutrient absorption directly facilitated biomass accumulation in Chinese cabbage, thereby improving its overall growth performance. Importantly, the enhanced nutrient availability, particularly N, also supports chlorophyll biosynthesis and photosynthetic activity, which are crucial processes for biomass accumulation.

Photosynthesis is a key step in the synthesis of OM and energy supply in plants [36]. Low concentrations of La can enhance photosynthesis by improving light energy absorption, electron transport, energy conversion efficiency, and chloroplast ultrastructure [37,38]. CSL contains various amino acids, including aspartic acid, glutamic acid, serine, glycine, and arginine, which can promote the synthesis of photosynthetic pigments by facilitating the production of key enzymes such as magnesium chelatase and protochlorophyllide oxidoreductase [39,40]. The present study demonstrated that CLa and CSL treatment enhanced the SPAD value, Pn, and Tr and increased photosynthetic pigment content in Chinese cabbage. Notably, the combined CLCS treatment produced superior results, increasing the SPAD, Pn and Tr by 2.6%, 294.8% and 40.1%, respectively (Figure 3a–c). In contrast, Ci and Gs decreased under the CLCS treatment (Figure 3d,e), a phenomenon seemingly inconsistent with the observed increases in Pn and Tr. A similar pattern was also reported in Liu’s study [41]. It was speculated that the observed decrease in Ci is attributable to La-enhanced leaf photochemical capacity: La likely improves electron transport and energy conversion efficiencies, which increases chloroplast CO_2_-use efficiency and accelerates consumption of CO_2_ [42]. The apparent contradiction between reduced Gs and elevated Tr may be explained by a CLCS-induced increase in stomatal density: a larger number of smaller stomata with reduced individual apertures would produce a lower measured Gs while sustaining or even enhancing overall gas exchange and transpiration [43]. Furthermore, the contents of chlorophyll a, chlorophyll b, carotenoids, and total chlorophyll were enhanced by 23.7%, 33.0%, 22.2%, and 25.0%, respectively, compared to CK (Figure 3f–i). Overall, these findings indicate that CLCS mitigates Cd stress in Chinese cabbage by simultaneously enhancing root architecture, nutrient uptake and photosynthetic efficiency, thereby leading to significantly improved biomass and yield. To further clarify the physiological mechanisms underlying this CLCS-induced growth improvement under Cd stress, its influence on Cd accumulation and antioxidant defense systems was examined subsequently.

### 2.3. Mechanisms Underlying Cd Detoxification

Cd^2+^ in the soil is absorbed by root cells through ion channels or transporter proteins on the plasma membrane, and then translocated to stems and leaves via xylem-localized transporters or as Cd-organic acid chelates [44]. Its accumulation in plant tissues triggers multiple forms of cellular damage, such as ROS accumulation and lipid peroxidation [45]. These oxidative damages are typically quantified by measuring H_2_O_2_ content and MDA content. The application of CLCS reduced shoot concentrations of Cd, H_2_O_2_, and MDA in Chinese cabbage by 18.1%, 26.4%, and 11.3%, respectively, compared to CK (Figure 4a–d). It was speculated in this study that CLCS alleviates Cd toxicity in Chinese cabbage primarily through two mechanisms: reducing Cd translocation from roots to shoots and enhancing antioxidant enzyme activity. Studies have shown that La limits the xylem-to-phloem transfer of Cd by promoting its sequestration in plant nodes and downregulating Cd transporter expression on the plasma membrane, thereby reducing Cd translocation to the shoot [46,47,48]. It was also speculated that the CLa may mitigate Cd mobility and bioavailability through coordination of Cd^2+^ with the thiol (–SH) and amino (–NH_2_) groups of Cys, thereby forming stable or bridged ternary chelates. In some cases, Cd^2+^ may even substitute for La^3+^ in the chelate, leading to the formation of Cd-Cys chelates that further reduce the pool of free Cd^2+^. Consistently, the CLa treatment lowered shoot Cd concentration and decreased the translocation factor (TF) by 9% compared with CK. A significant increase in soil pH was observed following the CLCS treatment, with values rising from 5.31 to 5.75. At higher pH levels, Cd tends to form stable metal humates with the humic acids in CSL, thereby reducing its mobility in the soil [49,50]. The present results corroborate these findings, with CLCS significantly reducing the BCF value by 14.3% and showing a non-significant 12.1% decrease in TF, thereby indicating lower Cd accumulation and a potential decline in translocation capacity in Chinese cabbage (Figure 4e,f).

Given its role in reducing Cd translocation, the effect of CLCS on antioxidant enzyme activities was investigated further. Antioxidant enzymes play a vital role in protecting plants from oxidative stress induced by Cd exposure. SOD catalyzes the dismutation of superoxide (O_2_^−^) into H_2_O_2_, which is then broken down into H_2_O and O_2_ by CAT and POD [51,52,53]. Furthermore, GSH is an essential metabolite that serves roles in antioxidant defense. It not only serves as a precursor for phytochelatin synthesis but also directly reduces H_2_O_2_ to H_2_O [54]. Total increases of 34.8%, 20.5% and 15.5% in SOD, CAT and GSH were observed in the CLCS treatment, compared to CK (Figure 4g–i and Figure 5a). However, the POD activity of Chinese cabbage under CLa, CSL, and CLCS treatments decreased by 9.2%, 21.3%, and 20.7%, respectively, compared to CK. POD commonly utilizes products of the phenylpropanoid pathway, such as phenolics and lignin precursors, as electron donors [55]. Therefore, its activity is significantly influenced by the overall metabolic status of the system. It was hypothesized that CLCS shifts energy allocation towards processes related to growth and repair, rather than towards phenylpropanoid metabolism, resulting in a decrease in POD activity. Previous studies have demonstrated that La enhances the activities of antioxidant enzymes and regulates GSH metabolism in plants under Cd stress [56]. Furthermore, Li et al. reported that the CSL treatment significantly increased the activities of SOD and CAT in Chinese cabbage exposed to Cd stress [57]. The results of this study also showed that SOD and CAT activities were lower under the CLa or CSL treatments alone compared to the CLCS treatment, indicating that the synergistic effect of CLa and CSL was greater than their individual effects. In summary, CLCS mitigates Cd stress through a dual mechanism: restricting Cd translocation and enhancing antioxidant defenses. This combined action not only reduces Cd uptake and transport but also mitigates oxidative damage, providing a deeper explanation for the observed restoration of growth under Cd exposure. Furthermore, these physiological improvements may also be linked to changes in the soil microbial community induced by CLCS application, warranting further investigation.

### 2.4. Integrated Insights into CLCS-Mediated Cd Remediation Efficacy

To comprehensively evaluate the relationships between various indicators and Chinese cabbage growth, principal component analysis and correlation heatmaps were used. The radar chart was used to visualize the effects of different treatments on soil physicochemical properties and principal component analysis (PCA) was conducted to further assess how changes in soil properties influenced the plant height and root length of Chinese cabbage (Figure 5b,c). According to the results, the CLCS treatment led to significant increases in soil pH, EC, AP, and AN, along with a notable reduction in DTPA-Cd concentrations. PCA results showed that the first two components explained 70.4% of the total variance, with PC1 and PC2 accounting for 49.4% and 21%, respectively. The PCA biplot revealed that the CK and CLa treatments clustered on the negative side of PC1, whereas the CLCS and CSL treatments clustered on the positive side. Notably, the CLCS treatment significantly increased soil nutrient levels (AN, AP, and EK) and pH, which were accompanied by enhanced plant height and fresh weight. In contrast, soil organic matter content decreased under CLCS. These findings suggest that CLCS established a more nutrient-rich and alkaline rhizosphere environment, thereby promoting robust growth of Chinese cabbage.

Correlation analysis highlighted the interdependence of soil nutrients, physiological status, and growth (Figure 5d). Aboveground fresh weight was strongly positively correlated with soil AP, EK, pH, EC and CAT activity, all factors indicative of improved nutrient status and stress resilience. In contrast, fresh weight correlated negatively with leaf MDA content, a marker of oxidative stress. Similarly, plant height tracked positively with both biomass and soil P. These multi-factor associations imply that higher nutrient supply and enhanced antioxidant defenses jointly drive growth.

### 2.5. CLCS-Induced Modulation of Rhizosphere Microbiota and Associated Functional Implications

Rhizosphere microbiota serve as critical mediators in the soil–plant system, simultaneously responding to environmental perturbations and modulating plant growth as well as heavy metal mobilization and uptake [58,59]. The rhizosphere microbial community composition obtained from 16S rDNA sequencing results exhibited that the number of OUTs identified in the CK, CLa, CSL and CLCS soils averaged 6904; 7839; 7889; and 8100, respectively. Proteobacteria, Actinobacteriota, Acidobacteriota, Gemmatimonadota, Chloroflexi, Bacteroidota, Planctomycetota, Patescibacteria and Firmicutes were the dominant phyla in the soil, accounting for 97.1% of the total reads (Figure 6a,b). These bacterial groups are commonly found in Cd-contaminated soils and are considered part of the core soil microbiota, suggesting their potential role in mitigating Cd toxicity [60]. Compared to the CK treatment, the CLCS treatment showed increased relative abundances of Actinobacteriota (from 12% to 13%). Members of the phylum Actinobacteriota play important functional roles in soil ecosystems, contributing to plant growth and stress resilience via a variety of mechanisms. They secrete phytohormones such as gibberellins and indole-3-acetic acid to promote plant growth and release extracellular polymers and enzymes to solubilize insoluble soil nutrients, enhancing nutrient uptake. Moreover, their cell surfaces contain numerous active sites capable of adsorbing Cd, contributing to effective Cd detoxification [60,61,62,63]. In contrast, Acidobacteriota, which thrive in acidic soils, showed a decreased abundance under CLCS treatment (from 12% to 11%), likely due to the associated increase in soil pH [64]. The relative abundances of other bacterial taxa remained largely unchanged, indicating that CLCS treatment preserved the overall composition of the rhizosphere microbiota.

The Richness, Shannon and Chao1 indices were used to measure the α-diversity of the rhizosphere microbiota (Appendix A). Richness reflects the observed number of species within a community, providing a direct measure of species count. Chao1 is an estimator of species richness that accounts for rare taxa, thus offering a more accurate prediction of the true species number. In contrast, the Shannon index incorporates both species richness and evenness, representing not only how many species are present but also how evenly individuals are distributed among them [65]. All three indices exhibited a decreasing trend under different treatments, although the differences were not statistically significant. Specifically, the CLCS treatment reduced the Richness index from 1838 in the control to 1808, the Shannon index from 8.90 to 8.85, and the Chao1 index from 1859 to 1857. Overall, although CLCS slightly decreased species counts, it did not cause a measurable disruption of the overall microbial community structure based on these α-diversity metrics.

Linear discriminant analysis effect size (LEfSe) revealed significant alterations in the abundance of specific bacteria in the CLCS treatment soil, as compared to the soil in the CK treatment (Figure 6c,d). In contrast to the CK treatment, the CLCS treatment altered the abundances of eight bacterial taxa, with Actinobacteria, Saccharimonadaceae, and *Micrococcus* being significantly enriched. *Micrococcus* mitigates Cd toxicity by reducing Cd^2+^ bioavailability through extracellular immobilization mechanisms, and simultaneously enhances plant tolerance by activating antioxidant enzymes (CAT, SOD), thereby maintaining cellular redox homeostasis and promoting plant growth [66,67]. These results suggest that the CLCS treatment enriches beneficial taxa, which contribute to Cd immobilization and enhanced plant stress tolerance.

The present study has shown that the CLCS treatment restores Chinese cabbage growth in Cd-contaminated soil by stabilizing the core soil microbiome—thereby preserving essential ecological functions—and by selectively enriching Cd-immobilizing and plant growth-promoting bacteria to strengthen rhizosphere processes involved in Cd detoxification, nutrient supply, and plant growth promotion.

In conclusion, the application of CLCS is an effective strategy to alleviate Cd toxicity in Chinese cabbage grown in contaminated soils. It not only reduces Cd translocation within the plant but also offers notable agricultural benefits. However, the resource utilization of CLa still requires further research to optimize its preparation and assess its performance in large-scale field applications. The synergistic use of CLa and CSL proposed in this study holds greater value and may enhance the commercial potential of CLa. Further field studies are needed to evaluate the applicability of the CLCS system across different crops and soil types.

## 3. Materials and Methods

### 3.1. Material Synthesis and Characterization

The CLa was synthesized through an ultrasonication-assisted hydrothermal (UAH) method. In summary, the Cys and La(NO_3_)_3_ (3:1 *w*/*w*) were dissolved in deionized water and simultaneously stirred magnetically while adjusting the pH to 7 using 0.1 mol L NaOH. This mixture was centrifuged to remove the precipitate. The supernatant was then subjected to ultrasonic treatment (QSONICA probe sonicator, 20 kHz, 100% amplitude) for 10 min, followed by hydrothermal processing in a 70 °C water bath for 9 h. After freeze-drying, the solid product was collected and ground into a white powder. The crystalline structure of CLa was characterized using an X-ray diffractometer (XRD, Rigaku Ultima IV, Tokyo, Japan) with Cu-Kα radiation (40 kV, 40 mA). Scans were performed over a 2 *θ* range of 5–60° with a step size of 0.02°. Functional group analysis was performed using a Nicolet iS20 FTIR spectrometer (Thermo Fisher Scientific, WA, USA). Samples were prepared as KBr pellets and scanned across the spectral range of 4000–400 cm^−1^ with a resolution of 4 cm^−1^.

CSL was obtained from Shandong Haochen Biotechnology Co., Ltd., Zaozhuang, China with the following physicochemical properties: pH 4.50, electrical conductivity (EC) 29.7 mS cm^−1^, organic matter (OM) 200 g L^−1^, total nitrogen (TN) 30.7 g L^−1^, total phosphorus (TP) 17.4 g L^−1^, total potassium (TK) 52.4 g L^−1^, amino acid 80.0 g L^−1^, organic acid 298 g L^−1^ and saccharide 100 g L^−1^.

### 3.2. Experimental Setup

Cd-contaminated soil was collected from agricultural fields in Xiangtan City, Hunan Province (27°94′41″ N, 112°97′08″ E). Visible impurities (e.g., gravel, plant residues) were manually removed prior to analysis. The soil exhibited the following characteristics: pH 5.31, EC 292 μS cm^−1^, OM 41.6 g kg^−1^, alkaline nitrogen (AN) 129 mg kg^−1^, available phosphorus (AP) 29.7 mg kg^−1^, exchangeable potassium (EK) 219 mg kg^−1^, total Cd 2.15 mg kg^−1^, and bioavailable Cd 1.51 mg kg^−1^.

Chinese cabbage (*Brassica rapa* subsp. *pekinensis* Cv. ‘Jing Cui 60’) was cultivated in a controlled greenhouse under the following conditions: 16/8 h light/dark photoperiod, 70–75% relative humidity, 700 ± 20 µmol m^−2^ s^−1^ photosynthetically active radiation (PAR), and 26/20 °C day/night temperatures. The experimental design included four treatments: (i) Control (CK, deionized water), (ii) CLa, (iii) CSL, and (iv) CLCS (CLa + CSL) (Table 2). Each treatment was replicated three times in a completely randomized design. Surface-sterilized seeds were sown in commercial substrate (Pindstrup, Denmark, pH 5.5–6.6, OM 177 g kg^−1^, CEC 26.9 mol kg^−1^, inorganic N 29.9 mg kg^−1^, P 190 mg kg^−1^, and K 9.42 mg kg^−1^). Seedlings at the two-true-leaf stage were transplanted into pots (14 cm top diameter × 10 cm bottom diameter × 16 cm height) containing 800 g of soil. Soil moisture was maintained at 65% of water-holding capacity through daily irrigation. Starting 7 days after transplantation, 100 mL of treatment solution was applied to the root zone at 7-day intervals for three consecutive weeks. A 30-10-10 NPK compound fertilizer (0.5 g kg^−1^ soil) was applied on day 15. After 30 days of growth, plants were harvested and separated into shoots and roots, and rhizosphere soil samples were collected and stored at −80 °C for subsequent analysis.

### 3.3. Plant Growth and Physiological Analyses

#### 3.3.1. Plant Growth Parameters

Plant samples were divided into above and underground parts. The adhering soil was rinsed off with potable tap water, followed by three rinses with deionized water. The root systems were then scanned using a Microtek ScanWizard scanner (Zhongjing Tech, Shanghai, China), and root morphological parameters—including total length, total surface area, total volume, average root diameter, and total number of root tips—were analyzed using the WinRHIZO 2007 root analysis system. Root activity was assessed via the triphenyltetrazolium chloride (TTC) reduction assay [68]. Fresh root samples were incubated in a solution containing 0.4% TTC and phosphate buffer (pH 7.0) at 37 °C in the dark for 2 h, and the reaction was terminated with sulfuric acid. The resulting triphenylformazan (TPF) was extracted with ethyl acetate, and the absorbance was measured at 485 nm to quantify root activity.

Photosynthetic pigment and photosynthetic parameters (Pn, Tr, Ci, Gs) were measured as follows: Photosynthetic pigment quantification was conducted on the third fully expanded leaf from the apex collected at harvest. Fresh leaf tissue (0.1 g) was homogenized in 10 mL of extraction solvent (80% acetone in absolute ethanol, 1:1 *v*/*v*) using a pestle and mortar. The homogenate was filtered, and 1 mL filtrate was combined with 2 mL acetone for spectrophotometric analysis. Chlorophyll a, chlorophyll b, and carotenoid concentrations were determined by measuring absorbance at 663 nm, 646 nm and 470 nm, respectively, on a spectrophotometer [69]. Net photosynthetic rate (Pn), intercellular CO_2_ concentration (Ci), stomatal conductance (Gs), and transpiration rate (Tr) of the leaves were measured under steady-state conditions (09:00–11:00 AM) using a portable photosynthesis system (LD-GH80, Leander Tech, Weifang, China) [70].

Plant tissue nutrient content was determined through acid digestion and spectrophotometric methods [71]. Aboveground dry samples of Chinese cabbage were digested with H_2_SO_4_-H_2_O_2_ (5:1, *v*/*v*), diluted to 50 mL, filtered, and analyzed. Automated Kjeldahl distillation-titration (K9840) was used to measure N, AP was measured by NaHCO_3_ extraction-Mo-Sb spectrophotometry, and K by flame photometry with LiNO_3_ as internal standard.

#### 3.3.2. Assessment of Oxidative Stress Markers in Plants

To quantify oxidative stress in Chinese cabbage, two key biomarkers were analyzed: malondialdehyde (MDA) content as an indicator of lipid peroxidation, and hydrogen peroxide (H_2_O_2_) concentration as a measure of ROS accumulation. The MDA content was determined according to the thiobarbituric acid reactive substances (TBARS) method [72]. Briefly, 200 mg of tissue were homogenized in 0.1% trichloroacetic acid (TCA), centrifuged at 10,000× *g* for 15 min, and 1 mL of the supernatant was reacted with 20% TCA and 0.5% TBA. After heating at 95 °C for 30 min, followed by cooling on ice, absorbance was measured at 532 and 600 nm, and the MDA concentration was calculated using an extinction coefficient of 155 mM^−1^ cm^−1^. H_2_O_2_ levels were determined following the peroxidase-coupled assay [73]. About 200 mg of tissue was extracted in 0.1% TCA, centrifuged at 12,000× *g* for 15 min, and the supernatant was incubated with potassium phosphate buffer (pH 7.0), 4-aminoantipyrine, phenol, and peroxidase (POD). Absorbance was recorded at 505 nm, and the H_2_O_2_ concentration was calculated from a standard curve.

#### 3.3.3. Analysis of Antioxidant Enzyme Activities

The crude enzyme extract was prepared by homogenizing seedlings in 50 mM phosphate buffer (pH 7.0) at 4 °C, followed by centrifugation at 15,000× *g* for 10 min. The subsequent supernatant was used as the crude extract. Superoxide dismutase (SOD) activity was determined via the riboflavin-nitroblue tetrazolium (NBT) method [74]. The 3-mL reaction mixture contained 50 mM phosphate buffer (pH 7.8), 13 mM methionine, 75 μM NBT, 2 μM riboflavin, 0.1 mM EDTA, and 0–150 μL of enzyme extract. After the addition of riboflavin, the mixture was exposed to light for 15 min, and the absorbance was measured at 560 nm. Catalase (CAT) activity was measured following Aebi and Lester [75]; the assay mixture contained 2 mL leaf extract, 50 mM potassium phosphate buffer (pH 7.0) and 10 mM H_2_O_2_. CAT activity was calculated at an absorbance wavelength of 240 nm. POD activity was determined via the guaiacol oxidation method [76]. The reaction mixture (3 mL) contained 50 mM phosphate buffer (pH 7.0), 20 mM guaiacol, 40 mM H_2_O_2_, and enzyme extract. The increase in absorbance at 470 nm was recorded to calculate POD activity. Glutathione (GSH) was measured using a regent kit (Elabscience Biotechnology Co., Ltd., Wuhan, China). All assays were performed in triplicate and temperature was maintained at 25 ± 0.5 °C during measurements.

### 3.4. Cadmium Analysis

Dried plant and soil samples were homogenized and sieved through a 0.15 mm mesh. For Cd analysis, 0.25 g aliquots were digested with 8 mL concentrated HNO_3_ (69%) using the following protocol: (i) Samples were pre-digested overnight at room temperature, (ii) complete digestion was achieved by heating at 120 °C until clear solutions were obtained, and (iii) digestates were cooled and diluted to 25 mL with ultrapure water. Cd concentrations were determined using flame atomic absorption spectrometry (FAAS, ZCA-1000SF8, Zhonghe Cetong, Beijing, China). Two key indices were calculated to evaluate Cd translocation and accumulation [77]:$$Transfer\;Factor\left(TF\right)=\frac{{Cd}_{shoot}+{Cd}_{leaves}}{{Cd}_{root}}$$$$Bioconcentration\;Factor\left(BCF\right)=\frac{{Cd}_{plant}}{{Cd}_{soil}}$$
where Cd_shoot_, Cd_leaves_, Cd_root_ and Cd_soil_ represent the Cd concentration (mg kg^−1^) in shoot, leaves, root, and initial soil, respectively.

### 3.5. Microbial High-Throughput Sequencing

For each treatment, soil adhering to the roots was collected, and approximately 10 g of fresh rhizosphere soil was weighed and stored at −80 °C prior to DNA extraction. Rhizosphere microbiota DNA was extracted using a Soil DNA Kit (Tiangen Biotechnology Co., Ltd., Beijing, China) according to the manufacturer’s instructions. 16S rDNA genes of distinct regions (V3–V4) were amplified using a pair of primers, 338F (50′-ACTCCTACGGGAGGCAGCA-30′) and 860R (50′-GGACTACHVGGGTWTCTAAT-30′) [78]. The bacterial communities were analyzed on the BMKCloud (www.biocloud.net) (accessed on 1 March 2025).

### 3.6. Statistical Analysis

All experimental data were processed and analyzed using established statistical methods to ensure robust interpretation of results. Mean values and standard deviations were performed using IBM SPSS Statistics 27. For crystalline phase identification from X-ray diffraction patterns, MDI Jade 9 software was employed with reference to standard patterns in the Powder Diffraction File database (PDF-4). Data visualization and figure generation were conducted using Origin 2025 software (OriginLab Corporation, Northampton, MA, USA) with appropriate statistical annotations. The R platform (version 4.3.1) was used for all Rhizosphere Microbiota statistical calculations, and the ggplot2 package was used to create the graphs. Based on the information from these OTUs, Vegan package was utilized to calculate the community Alpha diversity (Shannon, Chao and Richness indices). Statistical significance between experimental groups was determined through one-way analysis of variance (ANOVA) implemented in SPSS Statistics 27.0 (IBM, Armonk, NY, USA). All statistical tests were conducted with three biological replicates per treatment group, and Duncan’s multiple range test was used for multiple comparisons (*p* < 0.05) across treatments.

## 4. Conclusions

This study demonstrates that developing a novel rare earth-amino acid chelate enhances the bioavailability of La for agricultural use and confirms the potential of CLCS to alleviate Cd toxicity in Chinese cabbage. CLCS promotes plant growth under Cd stress through an integrated mechanism involving both direct plant responses and indirect soil-microbial interactions. At the plant level, CLCS improves root architecture, nutrient uptake, photosynthetic pigment synthesis, photosynthetic efficiency, antioxidant defense, and restricts Cd translocation. At the soil level, CLCS increases pH and nutrient availability while it selectively enriches the rhizosphere microbiome associated with plant growth promotion and Cd immobilization. Combined, these processes form a synergistic soil–microbe–plant feedback loop that mitigates Cd toxicity and enhances growth.

## Figures and Tables

**Figure 1 plants-14-03040-f001:**
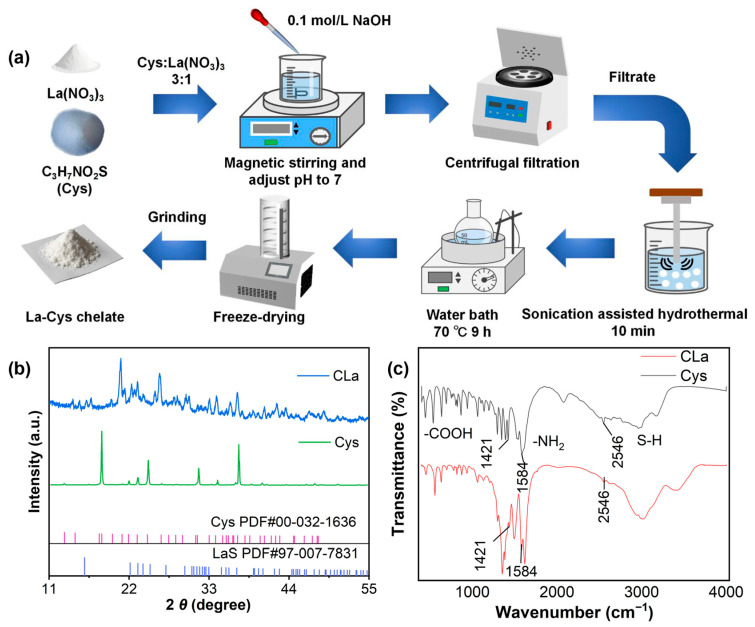
Schematic illustration of the synthesis and characterization of CLa. Synthesis flowchart (**a**), XRD pattern of CLa and Cys (**b**), FTIR spectrum of Cys and CLa (**c**).

**Figure 2 plants-14-03040-f002:**
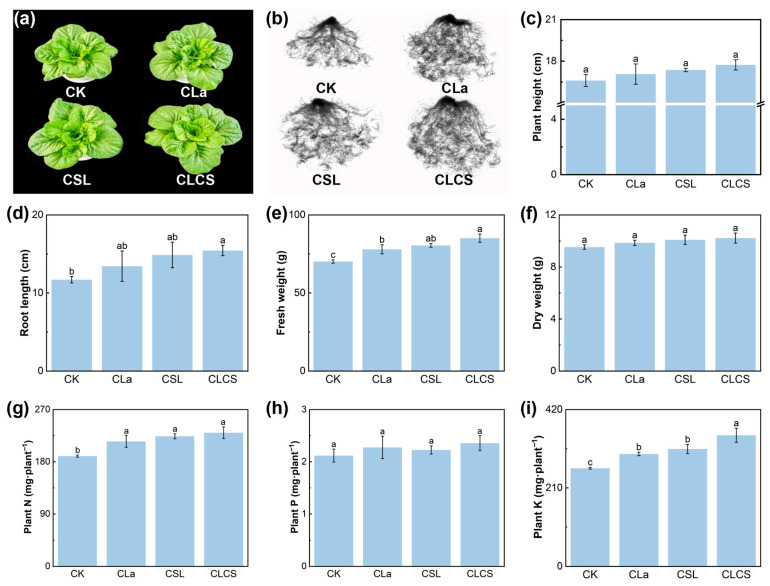
Effect of different treatments (CK, with deionized water; CLa, lanthanum-cysteine chelate; CSL, corn steep liquor; CLCS: the combined treatment of lanthanum-cysteine chelate and corn steep liquor) on plant growth (**a**), root growth (**b**), plant height (**c**), root length (**d**), shoot fresh weight (**e**), shoot dry weight (**f**), N content (**g**), P content (**h**), K content (**i**) of Chinese cabbage under Cd stress. Different lowercase letters indicate significant differences among treatments (*p* < 0.05; n = 3).

**Figure 3 plants-14-03040-f003:**
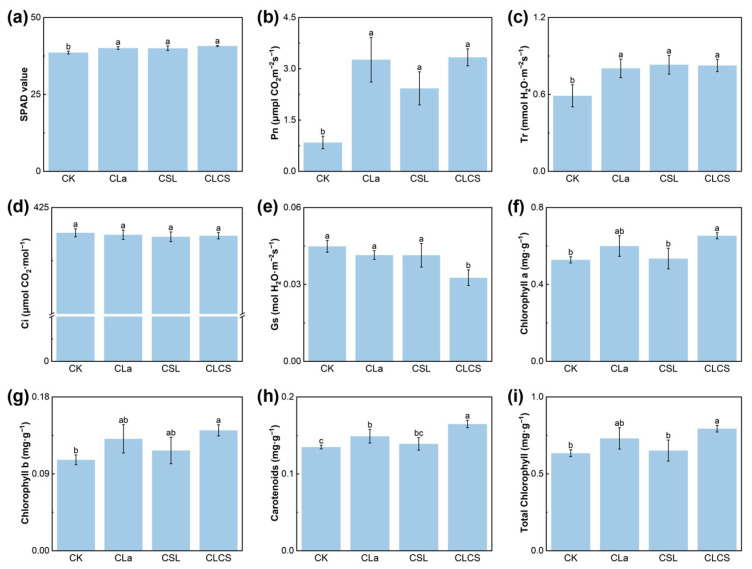
Effect of different treatments (CK, with deionized water; CLa, lanthanum-cysteine chelate; CSL, corn steep liquor; CLCS: the combined treatment of lanthanum-cysteine chelate and corn steep liquor) on photosynthesis, SPAD (**a**), Pn (**b**), Tr (**c**), Ci (**d**), Gs (**e**), chlorophyll a (**f**), chlorophyll b (**g**), carotenoids (**h**), total chlorophyll (**i**). Different lowercase letters indicate significant differences among treatments (*p* < 0.05; n = 3).

**Figure 4 plants-14-03040-f004:**
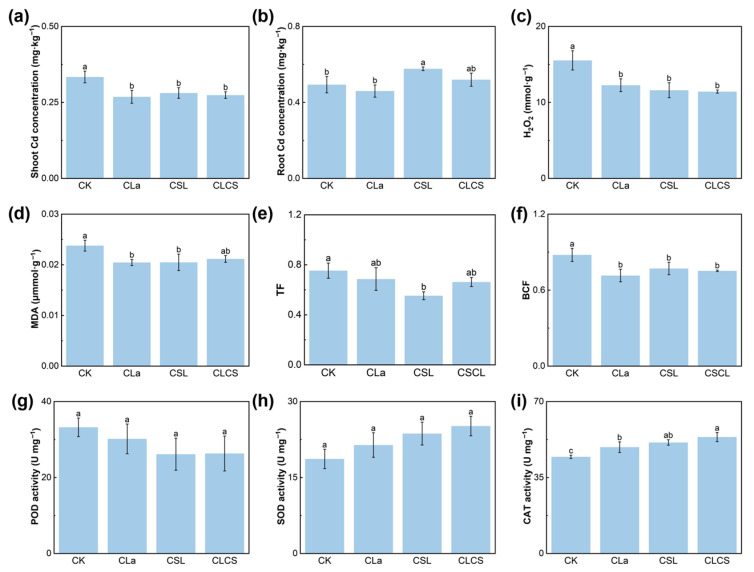
Effect of different treatments (CK, with deionized water; CLa, lanthanum-cysteine chelate; CSL, corn steep liquor; CLCS: the combined treatment of lanthanum-cysteine chelate and corn steep liquor) on Cd concentration in shoot (**a**), Cd concentration in root (**b**), H_2_O_2_ content (**c**), MDA content (**d**), TF (**e**), BCF (**f**), POD activity (**g**), SOD activity (**h**), CAT activity (**i**) of Chinese cabbage under Cd stress. Different lowercase letters indicate significant differences among treatments (*p* < 0.05; n = 3).

**Figure 5 plants-14-03040-f005:**
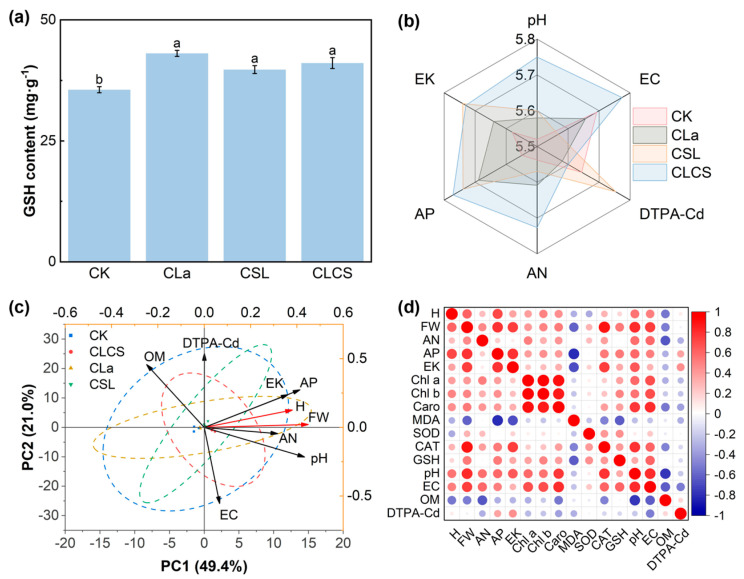
Effect of different treatments (CK, with deionized water; CLa, lanthanum-cysteine chelate; CSL, corn steep liquor; CLCS: the combined treatment of lanthanum-cysteine chelate and corn steep liquor) on GSH content (**a**). The radar chart of soil physicochemical properties (**b**). PCA of Chinese cabbage growth and soil physicochemical indicators (**c**). The correlation analysis between Chinese cabbage growth, photosynthetic pigments, antioxidant enzyme activities and soil physicochemical indicators (**d**). Different lowercase letters indicate significant differences among treatments (*p* < 0.05; n = 3).

**Figure 6 plants-14-03040-f006:**
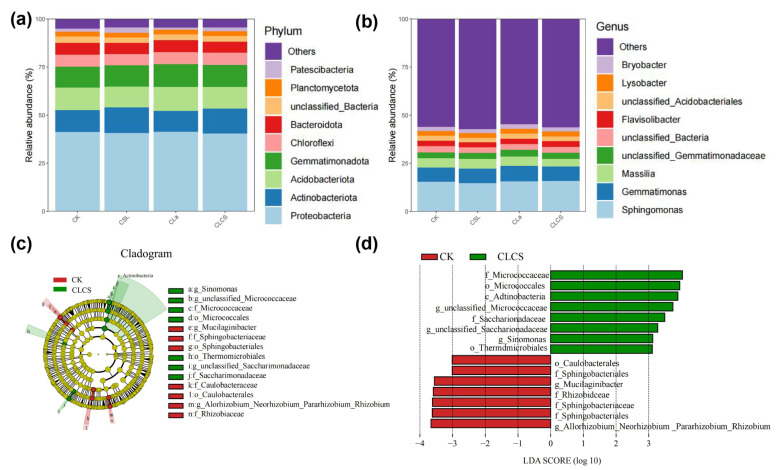
Effect of treatment on the rhizospheric microbial community in soil. Taxonomic composition at phylum level (**a**), taxonomic composition at genus level (**b**), LEfSe-derived cladogram of differentially enriched phylogenetic clades (**c**), linear discriminant analysis (LDA) score histogram showing effect sizes of discriminative features (**d**).

**Table 1 plants-14-03040-t001:** Effects of irrigating CLa and CSL on root system architecture.

Treatment	Surface Root Area (cm^2^)	Root Diameter (mm)	Root Volume (cm^3^)	Root Tips	Root Forks	Root Crossing	Root Activity (μg g^−1^ h^−1^)
CK	147 b	0.41 b	1.54 b	4230 b	13,359 b	4002 b	256.44 c
CLa	215 a	0.43 ab	2.10 b	4952 ab	17,941 ab	5368 ab	292 b
CSL	223 a	0.40 b	2.25 b	5764 a	20,484 a	6269 a	260.89 bc
CLCS	262 a	0.45 a	3.09 a	5998 a	22,477 a	6610 a	358.67 a

CK: with deionized water; CLa: lanthanum-cysteine chelate; CSL: corn steep liquor; CLCS: the combined treatment of lanthanum-cysteine chelate and corn steep liquor. Different lowercase letters indicate significant differences among treatments (*p* < 0.05; n = 3).

**Table 2 plants-14-03040-t002:** Experimental treatments.

Treatment	CLa (mg kg^−1^)	CSL (g kg^−1^)
CK	0	0
CLa	2.5	0
CSL	0	1.14
CLCS	2.5	1.14

CK: with deionized water; CLa: lanthanum-cysteine chelate; CSL: corn steep liquor; CLCS: the combined treatment of lanthanum-cysteine chelate and corn steep liquor.

## Data Availability

The original contributions presented in this study are included in the article/Appendix A. Further inquiries can be directed to the corresponding author.

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
