# Peer review of "Synergistic Lanthanum-Cysteine Chelate and Corn Steep Liquor Mitigate Cadmium Toxicity in Chinese Cabbage via Physiological–Microbial Coordination"

_plants, 2025, doi:10.3390/plants14193040_

Round 1
Reviewer 1 Report
Comments and Suggestions for Authors
This paper investigated the effects of CLa, CSL and CLCS on Chinese cabbage under cadmium stress, which has a certain degree of innovation. But this paper still has the following issues.
- For the “Materials and Methods” section, the current description is too simple. Please provide detailed experimental methods. In Figures 2-4, Does “n=3” represent measuring only 3 seedlings for each indicator?
- Lines 239, which result indicate “the role of microorganisms”.
- Figure 4i and Figure 5a, lowercase letters that indicate significant differences are incorrect.
- Lines 269-270, Please provide the pH data.
- Lines 272-274, these descriptions are not rigorous because there was no significant difference between TF of CK and CSCL.
- Lines 338-340, where is this result?
- Lines 350-358, Provide significance analysis results for Richness, Shannon and Chao1 indices.
- Why do CLa and CSL not have a significant effect when applied separately but have a significant effect when treated together (CLCS)?
- Where are the description of the count in Fig. S1-S3?
- For the “Conclusions” section, these conclusions are not rigorous and overly expanded. It is suggested to rewrite the conclusion section. For example, Lines 395-398, “…enhancing Cd sequestration in roots and downregulating Cd transporter expression.” Currently, all the results in the article do not support these conclusions.
Reviewer 2 Report
Comments and Suggestions for Authors
The research is scientifically relevant, well written, and well structured. I only noticed that sometimes 'figure' is written in full and sometimes abbreviated as 'fig.' within parentheses. Please adjust according to the journal’s guidelines or maintain a consistent format. The quality of Figure 6c and d should be improved, as the internal labels are not legible even when enlarged.
Reviewer 3 Report
Comments and Suggestions for Authors
In this study the authors evaluate a dual-component remediation strategy combining lanthanum–cysteine chelate and corn steep liquor to alleviate Cd toxicity in Chinese cabbage. The results of this research can be important for the cultivation of this species in regions of China with Cd polluted soils. This extensive study, involved a large volume of morphological, physiological, biochemical and microbiological measurements and analyses. However, in order to be published I consider that the manuscript needs certain improvements. All detailed comments and suggestions can be found bellow:
-I recommend you to arrange the keywords alphabetically.
- In the Introduction section there are no information regarding the Chinese cabbage, and its reaction to Cd pollution/stress.
-In M & M section, add reference(s) to the formulas for TF and BCF.
- In subsection 2.6 indicate what statistical (post-hoc) test (LSD, Duncan, Tukey, etc) was used to compare treatments
-According to the journal’s requirements (Research Manuscript Sections), you must split the text from section 3 (Results and Discussion) in two separate sections (3. Results; 4. Discussion).
- Remove „P < 0.05” from the main text. The significance of the differences between treatments is already (properly) presented based on letters in tables and figures.
- In the main text replace “Fig.” with Figure.
- Insert the figures/tables close to its associated text. For instance, Figure 3 along with Table 2 must be inserted after the line 225.
- The presentation of the results is done briefly, and must be improved. For instance, the text associated to the 14 parameters from Figure 2 and Table 2, along with some discussions is only 17 lines long (line 207-225).
-In the footer of Table 2 insert „Different lowercase letters indicate significant differences among treatments”.
- Considering that the values ​​for chlorophyll (a, b, total) are presented, I consider that the SPAD values ​​are not necessary. Also, it is observed that the SPAD values are not correlated with the other chlorophyll values. In the text associated to data from Figure 3 there are no comments regarding the effect of different treatments on Ci. Gs and SPAD.
-In the main text there are no information/comments regarding the data from Figures S1, S2, S3. Also, the seven treatments from Figures S1-3, are not presented (concretely) in the M&M section. The comments from lines 106-107 can be developed and transferred to Results section.
- In the title of Figure S4, you must explain the data (indices) associated to graph a (Richness), b (Shannon ) and c (Chao1).
- The data and conclusions associated to PCoA plot of bacterial communities (Figure S4d), cannot be considered relevant, given that PC1 (8.85%) and PC2 (7.52%) explain cumulatively only 16.37% from the total variation.
- For a better highlighting of the obtained results and considering that the data were processed using ANOVA, I consider that it is necessary to present (in the supplementary materials) the results of ANOVA (MS, F value) for each parameter/compound.
-The Discussion part (section) can be extended/improved, given that the actual size is approximately 45-50 lines.
Round 2
Reviewer 3 Report
Comments and Suggestions for Authors
Given that the authors have considered (most of) the received comments/suggestions, the new version of the manuscript is significantly improved. As such, I appreciate that the article meets the requirements of the journal and can be accepted in present form.